# DroneDreamer: Multi-View Low-Altitude World Model with Adaptive Control

## Abstract

Recent advancements in world models have made significant strides in the fields of autonomous driving and robotic motion, showing great potential as the foundation for general artificial intelligence. Although DiT with 3D VAE has made significant progress in this field, existing models have yet to be effectively applied to low-altitude scenarios, still facing challenges such as difficulty in acquiring conditions, unstable sampling, and data scarcity, which render current methods ineffective. In this paper, we introduce a new research problem: Low-Altitude World Models(LAWM), focusing on the applicability of world models in low-altitude scenarios. To address these issues, we propose DroneDreamer, a novel LAWM that incorporates an adaptive viewpoint control mechanism and an image style domain adaptation technique, enabling multi-view conditional generation with limited conditions. Additionally, we construct a drone flight dataset collected from a simulated modeling environment and a data processing pipeline, along with a progressive training strategy to ensure the accuracy of control capabilities and style adaptation, as well as generalizability. Consequently, our model achieves a 58% improvement in image generation capabilities on the drone flight dataset compared to specialized models, and a 63% improvement in multi-view consistency compared to combined baseline models. We believe DroneDreamer can provide a foundational world model for the low-altitude UAV domain, benefiting low-altitude navigation tasks, data generation, flight prediction, and inspiring valuable applications.Our code is available at https://anonymous.4open.science/r/DroneDreamer-B629

## 1 Introduction

In recent years, world models have demonstrated significant success (Yu et al., 2025; Li et al., 2025; OpenAI, 2024), proving their potential as foundations for general artificial intelligence. Multi-view video generation enhances world models' ability to perceive and understand environments in tasks like autonomous driving (Zhou et al., 2024b; Gao et al., 2024c; Zhou et al., 2025b) and robot motion control (Liu et al., 2025). The low-altitude UAV domain, marked by challenges such as data scarcity and complex environments, has seen attempts to apply world models (Zhao et al., 2025). Multi-view low-altitude world models are thus crucial for expanding the scope of world models and improving their ability to manage complex environments.

Current research on multi-view generation mainly focuses on robot motion (Sakagami et al., 2023) and autonomous driving (Guan et al., 2024), both requiring complex control conditions. While robot motion emphasizes manipulation and indoor modeling, autonomous driving centers on lane trajectories and driving behaviors. These approaches face challenges in complex, multi-dimensional low-altitude environments, where dataset scarcity and the difficulty of obtaining traditional control conditions lead to uncontrollable videos and discrepancies with real-world environments. Recent efforts have attempted to separate background from objects (Zhou et al., 2024b), and improve video resolution and generation length (Gao et al., 2024c; Ruiyuan Gao, 2025). However, these models are limited by their domains and the complexity of their control conditions.

This paper introduces a novel research problem:Low-Altitude World Model (LAWM). The goal is to establish a world model capable of perceiving, understanding, and generating complex low-altitude

---

*Corresponding author.

**Input prompt**:On a sunny day in a green suburban park adjacent to a road, the drone moves steadily forward ...

Current Observation

Frame 1  Frame 6

Frame 11  Frame 16

Figure 1: In low-altitude environments, DroneDreamer can predict trajectory-consistent multi-view flight videos based on the current front-view observation. The six viewpoints are evenly distributed around the UAV in a circular arrangement.

environments, with multi-view as a key perceptual ability. Our approach proposes a low-altitude world model with simple input conditions, controllable trajectory generation, omnidirectional multi-view perception, and output adaptation across various styles. Derived from world models (Ha & Schmidhuber, 2018) and navigation models (Bar et al., 2025), this model goes beyond video generation, positioning it as a comprehensive world model. It expands existing research by providing a platform for low-altitude interactive environments. A demonstration of the model's generated examples is shown in Figure 1.

Building on the LAWM concept, we propose DroneDreamer, a novel low-altitude multi-view world model. Extending the MagicDrive-V2 architecture (Ruiyuan Gao, 2025), we introduce input adaptive masking and image style adaptation mechanisms. The input adaptive masking mechanism enables adaptation to varying conditions by detecting inputs and concatenating denoising layers, addressing missing control conditions in low-altitude environments. The image style adaptation mechanism utilizes class-wise AdaIN from image style transfer (Chigot et al., 2025) to handle different style outputs and compensate for missing real data. We also constructed a six-view UAV flight dataset with diverse styles, trajectory, and semantic segmentation annotations, formatted in nuScenes (Caesar et al., 2020).

We conducted evaluations on the constructed low-altitude UAV flight dataset and compared it with three widely accepted models, including both end-to-end models and combined models. Our experiments show that DroneDreamer outperforms end-to-end multi-view models, achieving improvements in the quality of various style outputs and multi-view consistency. Compared to combined models, we achieve similar image quality with fewer parameters and faster inference speeds, while significantly surpassing existing models in multi-view consistency. Besides, DroneDreamer also surpasses other models in terms of generation stability. More importantly, DroneDreamer's strong historical frame control capabilities provide reliable predictive simulations for UAV navigation.

The key contributions of this work are as follows:

- We introduce low-altitude world models, expanding the research dimensions and application space of existing models.

- We propose the first low-altitude multi-view world model, addressing key challenges with *input adaptive masking* and *image style adaptation*, and construct a high-quality multi-view dataset for the low-altitude domain.

- Experiments show that our model outperforms others in multi-view consistency and provides reliable predictions, laying the foundation for future UAV navigation with our enhanced control capabilities.

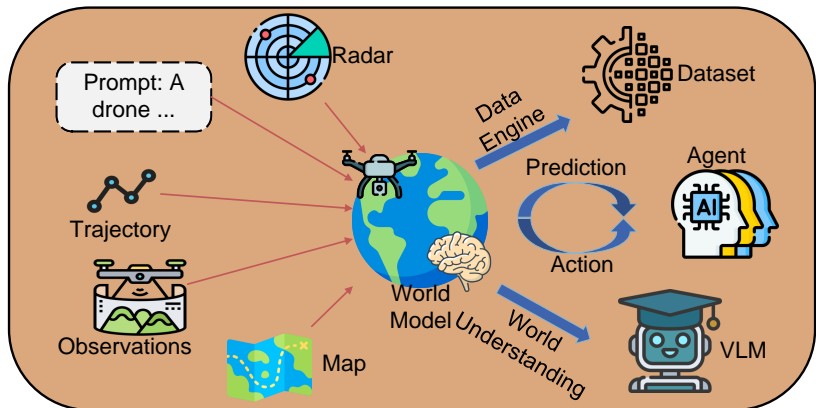

Figure 2: Illustration of the role and positioning of Low-Altitude world models

## 2 LOW-ALTITUDE WORLD MODEL

### 2.1 BACKGROUND

World models have been widely explored in recent years (Ding et al., 2025), with applications in robot motion prediction (Hu et al., 2025), edge data generation for autonomous driving (NVIDIA, 2025a), and text-to-video exploration (Yang et al., 2025). Recent studies, such as NWM (Bar et al., 2025) and Airscape (Zhao et al., 2025), attempt to apply world models in real-world settings. Our contribution lies in defining the role of world models in low-altitude environments and addressing their unique challenges, providing a new research direction.

Low-altitude world models apply to complex three-dimensional environments, where they perceive, interpret multimodal inputs, and generate environmental predictions, as shown in Figure 2. However, creating such models still faces significant challenges:

- Lack of aerial datasets. Training such a world model requires multi-view flight data from real-world environments along with corresponding camera angle annotations. Existing datasets are either focused on the autonomous driving domain or consist of single-view flight data.

- Missing control conditions in low-altitude environments. Existing multi-view video generation models rely on 3D modeling of the surrounding environment and BEV-Maps, which are limited to confined spaces or planar motion. In low-altitude scenarios, traditional control conditions are costly to obtain, and the top-down view cannot account for height variations.

- The distribution gap between base models and low-altitude world models. In multi-view video generation, existing open-source autonomous driving models focus on lane-keeping and vehicle driving behaviors. However, UAVs operate under more degrees of freedom, generating scenes that include on-site rotation, height variations, and combinations of multiple actions, making the generation more challenging.

To address these issues, we first construct a dataset of 350 scenes to train world models. We collect multi-view UAV flight data from a simulated environment and use large multimodal models to annotate the dataset with scene descriptions. To supplement data style, we first perform semantic segmentation on images using the EoMT (Kerssies et al., 2025) method, and then apply the style transfer model from CACTIF (Chigot et al., 2025) to reference real flight images for style adaptation. Subsequently, we design a three-stage training plan to train the model as a multi-view low-altitude world model.

### 2.2 PRELIMINARIES

**Diffusion Models** Denoising Diffusion Probabilistic Models (DDPMs) (Ho et al., 2020) are a family of generative models that define a forward noising process and a reverse denoising process. In the

forward diffusion process, Gaussian noise is gradually added to the original data $x_0$ over $T$ steps. In contrast to the forward diffusion process, the reverse denoising process is the procedure of iteratively removing noise from $x_T$ to recover the original data distribution $x_0$. In practical applications, we typically perform the diffusion and denoising processes in the latent space sampled from the VAE (Rombach et al., 2022). To train such a model, recent research mainly adopts the v-prediction method (Esser et al., 2024), training the model to predict the denoising speed and minimizing the MSE loss, the loss is defined as:

$$x_t = (1-t)x_0 + t\epsilon$$

$$\mathcal{L}_{diffussion} = \mathbb{E}_{\epsilon \sim \mathcal{N}(0,I)} \left\| \mathcal{V}_\theta(x_t, t) - \frac{1}{1-t}(x_t - \epsilon) \right\|_2^2$$

where $t \sim \text{lognorm}(0,1)$ is normalized time step and $\mathcal{V}_\theta$ is the model.

**Problem Formulation** This paper addresses the problem of low-altitude multi-view controllable video generation. Given a sequence of frame conditions $\{S_t\}, t \in \{0, \ldots, T\}$, the goal is to generate the corresponding low-altitude flight video from the latent variable $z \sim \mathcal{N}(0, \mathbf{I})$ under the given conditions, i.e., $\{I_{c,t}\} = \mathcal{F}(\{S_t\}, z_t)$, where $c \in \{0, \ldots, C\}$ represents C viewpoints. To achieve control over the generated video, we modify the traditional autonomous driving control conditions (Ruiyuan Gao, 2025; Gao et al., 2024b) by adding or removing conditions. Specifically, the frame control conditions $S_t = \{C, L, Tr_t, Seg_t, I_0\}$ include camera angle information $\{C\} = [R_c, t_c]$, where $L$ describes the entire flight motion scene, representing the overall flight environment and behaviors, $T_r$ represents the rotation and position changes relative to the first frame at each time step, $Seg$ captures the semantic information of different parts in the viewpoint, and the first-frame image, $I_0$, which can be an image from any moment and any viewpoint, offers a flexible control option for generating frames from any viewpoint. Therefore, the training objective of this paper is defined as:

$$\mathcal{L}_{con-dif} = \mathbb{E}_{\epsilon \sim \mathcal{N}(0,I)} \left\| \mathcal{V}_\theta(x_t, t \mid S_t) - \frac{1}{1-t}(x_t - \epsilon) \right\|_2^2$$

## 3 DroneDreamer Framework

### 3.1 Low-altitude Multi-view Dataset

We constructed a multi-view UAV flight dataset containing 365 urban scenes, with corresponding trajectory, text, and semantic segmentation annotations. The dataset was created using our custom pipeline, which incorporates semantic segmentation and style transfer for realistic data adaptation. Detailed descriptions of the dataset construction process and statistics can be found in Appendix A.2.

### 3.2 Overview of DroneDreamer

Figure 3 illustrates an overview of the model architecture. Building upon the MVDiT architecture of MagicDrive-V2 (Ruiyuan Gao, 2025), we retain the injection branch for control signals (NVIDIA, 2025a) and employ cross-attention mechanisms for text, trajectory and camera viewpoints (Rombach et al., 2022). Our model extends the 3D VAE + MVDiT framework, where input images are downsampled through the 3D VAE and subsequently undergo a denoising process via MVDiT.

However, the original primary control signals, such as the 3D bounding box and BEV-map, pose significant challenges in low-altitude environments. The 3D bounding box is difficult to obtain and highly complex, while the BEV-map, when transitioning from an autonomous driving planar environment to a more intricate 3D low-altitude environment, fails to fully represent complex height information. Therefore, we have removed these two controls and introduced a simple yet efficient input adaptive masking control, using visual information as a conditional control for generation. Further details are provided in Section 3.3.

Another challenge is that our data come primarily from flight scenes in a simulated environment, resulting in a domain gap between the data distribution and real-world environments. To address this issue, we have incorporated class-wise AdaIN and a main-view style cross-attention mechanism (Chigot et al., 2025) into the model, thereby reducing the distribution gap between generated images and real-world environments. Further details are provided in Section 3.4.

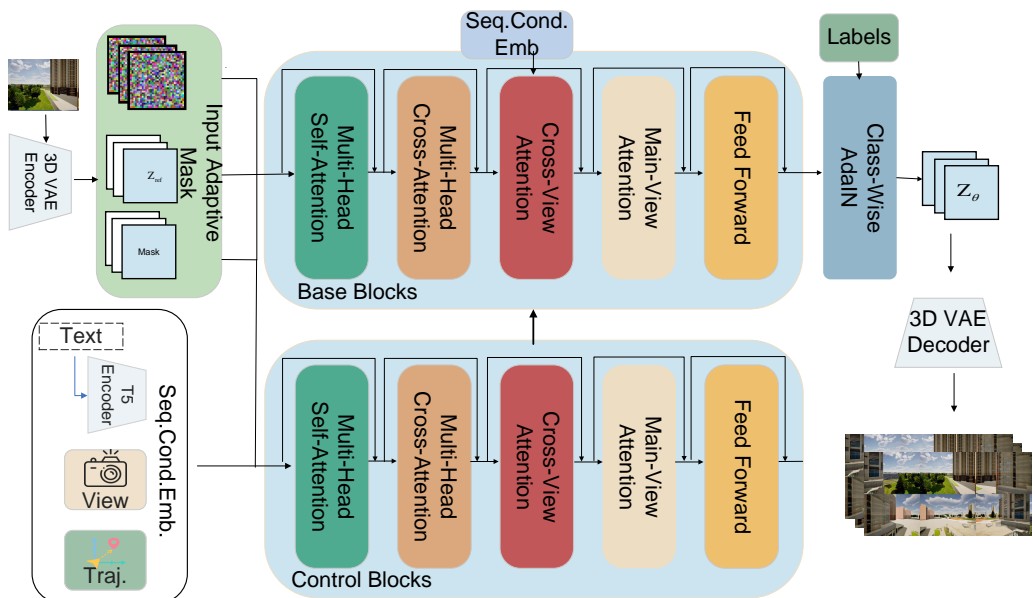

Figure 3: Overview of the DroneDreamer architecture. DroneDreamer takes multimodal information and current observations as input to generate multi-view predictive videos. We introduce an input adaptive masking mechanism and a style adaptation module to address the challenges of the low-altitude domain.

Furthermore, DroneDreamer employs a progressive training strategy, starting from simulated images to simulated images guided by the main view, and then to real-world images with main view guidance and style transfer, thereby enhancing the model's scalability and control capabilities. Additionally, out-of-frame extrapolation and adaptive control extrapolation are performed beyond the training setup (Section 3.5).

## 3.3 INPUT ADAPTIVE MASKING CONTROL MECHANISM

Transitioning from the original flat autonomous driving environment to a multi-degree-of-freedom low-altitude environment presents a key challenge: maintaining model stability and controllability in the absence of traditional control conditions. To address this limitation, we employ reference image injection to ensure the controllability of the generated video. The original MagicDrive-V2 base model (Ruiyuan Gao, 2025) does not support this functionality, and without traditional 3D bounding boxes and BEV-maps, it often suffers from inconsistent generation sampling and missing buildings. Since the reference image operates in a different dimensional space compared to the original control conditions, it should not be injected through the same control branch as the original control conditions.

To address this issue, inspired by the work in HyperMotion (Xu et al., 2025), we extend its latent space and manually designed mask to incorporate viewpoint control and reference frame control, utilizing an adaptive masking mechanism, as shown in Figure 3. When reference frames are provided, which can be from any viewpoint and for any duration, we first create an empty vector with the same dimensions as the video to be denoised. Then, the reference frames, encoded through the 3D VAE structure, are embedded into this empty vector to obtain the latent vector $L_{\mathbf{ref}} \in \mathbb{R}^{B \times NC \times C \times T \times H \times W}$. Simultaneously, a binary mask $M \in \mathbb{R}^{B \times NC \times 3 \times T \times H \times W}$ is constructed, where $M = 1$ at the positions where the reference is provided and $M = 0$ elsewhere. This mask explicitly controls the positions to be denoised and the positions where the reference is provided. Finally, the reference latent, mask latent, and denoised latent are concatenated along the channel dimension to form the final input to the denoising network:

$$L_{\text{input}} = \text{Concat}(L_{\text{noisy}}, L_{\text{ref}}, M_{\text{ref}}) \in \mathbb{R}^{B \times NC \times 35 \times T \times H \times W} \qquad (1)$$

### 3.4 STYLE ADAPTATION DESIGN

Our primary data is sourced from drone flight data collected in a simulated environment. There exists a domain gap between simulated and real-world data, such as differences in color distribution, surface details, etc. However, multi-view drone flight faces limitations like data scarcity, high annotation acquisition costs, and complex scenes. Applying style adaptation techniques to enhance the model's few-shot style adaptation capability is crucial. Recent works have demonstrated that diffusion models possess domain adaptation capabilities (Zhang et al., 2023), but post-processing techniques can significantly increase model parameters and inference time. Therefore, we integrate domain adaptation capabilities into the existing model through architectural modifications, rather than relying on post-processing.

We applied two methods that are commonly used in recent style transfer models. First, we add a main-view cross-attention layer to the original MVDiT block. When the latent undergoes attention computation in this layer, cross-attention is applied between the images from the other viewpoints and the main-view image as follows:

$$\text{Attention}(Q_{\text{ov}}, K_{\text{mv}}, V_{\text{mv}}) = \text{softmax}\left(\frac{Q_{\text{ov}} \cdot K_{\text{mv}}^T}{\sqrt{d}}\right) \cdot V_{\text{mv}} \tag{2}$$

Next, we add a class-wise AdaIN layer at the output of each denoising step in the model. To ensure that the image semantic labels and latents are aligned in dimensions, we sample the labels at the same scale $I_{label} \in \mathbb{R}^{B \times NC \times T \times H \times W}$. Specifically, the class-wise AdaIN layer is defined as follows:

$$\text{AdaIN}(I_{\text{ov}}, I_{\text{mv}}) = \sigma(I_{\text{mv}})\left(\frac{I_{\text{ov}} - \mu(I_{\text{ov}})}{\sigma(I_{\text{ov}})}\right) + \mu(I_{\text{mv}}) \tag{3}$$

where $\mu$ and $\sigma$ represent the mean and standard deviation calculated across the spatial dimensions for each channel and each semantic category. This design enables the transfer of texture and color information while ensuring that the semantic information of the image remains unchanged. In the absence of label input, we use a global AdaIN strategy.

### 3.5 THREE-STAGE MODEL TRAINING STRATEGY

To gradually transition the model to low-altitude scenes and improve controllability, we adopt a three-stage training strategy: starting with uncontrolled simulated flight scenes, transitioning to simulated flight scenes with main-view control, and finally using a small dataset of real-world style-transformed flight data with main-view control. This strategy is based on past experience with autonomous driving world model training: in controllable generation, the model prioritizes generation quality over learning controllability (Gao et al., 2024d).

In the first stage, we train using all simulated environment flight data with a relatively high learning rate, allowing the model to quickly transition into the generation domain. In the second stage, we introduce the input adaptive masking mechanism and a new fully connected layer, inputting the main-view images from the flight videos to train the control capabilities of reference frames. In the third stage, we freeze the original model parameters and only train the newly added main-view cross-attention layer, inputting a small number of style-transformed images along with semantic segmentation. This training strategy allows the model to generalize over the original reference viewpoints and time, while providing diverse input options.

## 4 EXPERIMENTS

### 4.1 EXPERIMENTAL SETUP

**Dataset and Baseline.** We performed validation on the low-altitude dataset we constructed, which includes multi-view drone flight data with various styles and flight behaviors. Our baseline models are divided into two categories: end-to-end models and composite models. The end-to-end model employs the MagicDrive-V2 fine-tuned on the low-altitude dataset. The composite models include Qwen-Image-Edit + CogVideoX-5B-I2V (Yang et al., 2025) and stable virtual camera(SVC) + CogVideoX-5B-I2V (Zhou et al., 2025a; Yang et al., 2025), which belong to the multimodal large

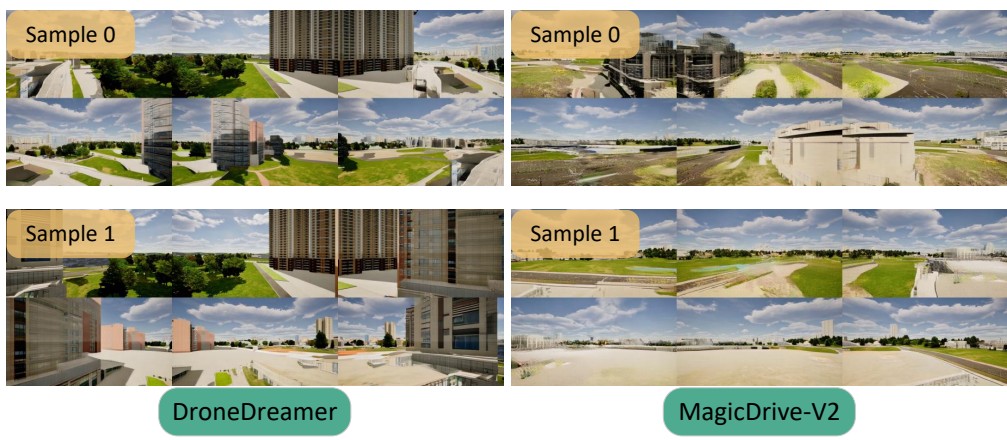

Figure 4: From top to bottom, we show the repeated sampling results of DroneDreamer and the baseline model.

| Model | Hybrid Styles | | | Realistic Styles | | | Simulated Styles | | |
|---|---|---|---|---|---|---|---|---|---|
| | FID ↓ | mIoU ↑ | FVD ↓ | FID ↓ | mIoU ↑ | FVD ↓ | FID ↓ | mIoU ↑ | FVD ↓ |
| Magicdrive-V2 | 173.02 | 0.150 | 1500.62 | 230.06 | 0.149 | 1619.90 | 172.51 | 0.149 | 1524.40 |
| Qwen+CogVideoX-i2v-5B | 51.08 | 0.125 | 1255.69 | **70.27** | 0.120 | 2342.03 | 59.67 | 0.128 | 973.68 |
| SVC+CogVideoX-i2v-5B | 66.81 | 0.123 | 1305.98 | 84.80 | 0.120 | 2607.53 | 81.34 | 0.124 | 892.35 |
| **DroneDreamer** | **73.31** | **0.238** | **463.94** | 83.62 | **0.238** | **516.21** | 138.53 | **0.237** | **493.84** |

Table 1: Evaluation of multi-style, six-view video prediction results of various world models in low-altitude environments.

model + video generation model and omnidirectional view generation model + video generation model, respectively.

**Evaluation Metrics.** We evaluate the realism and controllability of video and image street scene generation. For video generation, we adopt the same evaluation methodology as W-CODA2024 (Organizers, 2024), using FVD to assess both video quality and viewpoint consistency. For image generation, we adopt the metrics from MagicDrive (Gao et al., 2024c). Image quality is measured using FID, and the controllability is evaluated using mIoU. All metrics are evaluated on three styled image datasets, namely hybrid, realistic, and simulated styles.Experimental details are provided in the appendix A.3.

## 4.2 COMPARISON OF EXPERIMENTAL RESULTS

### 4.2.1 QUALITATIVE RESULTS

**Consistency Performance** Figure 5 illustrates the consistency issues faced by baseline models in low-altitude environments, as well as a comparison of generation results. The original baseline model exhibits insufficient viewpoint consistency. The generated viewpoint images from the baseline model are nearly identical, lacking rotational variation between viewpoints. In contrast, our DroneDreamer not only ensures image correspondence between multiple viewpoints but also maintains the spatial relationships of viewpoint rotations, generating more coherent multi-view videos.

**Comparison of Repeated Sampling Results** Figure 4 provides a comparison between our method and the baseline models, showing multiple samplings under fixed input conditions during drone flights. To validate the effectiveness of our control conditions, we conducted repeated sampling experiments on both the fine-tuned MagicDrive-V2 and DroneDreamer. The results demonstrate that our model maintains stability in the visible regions of the front view, while the baseline model exhibits significant variation, with entirely different scene compositions.

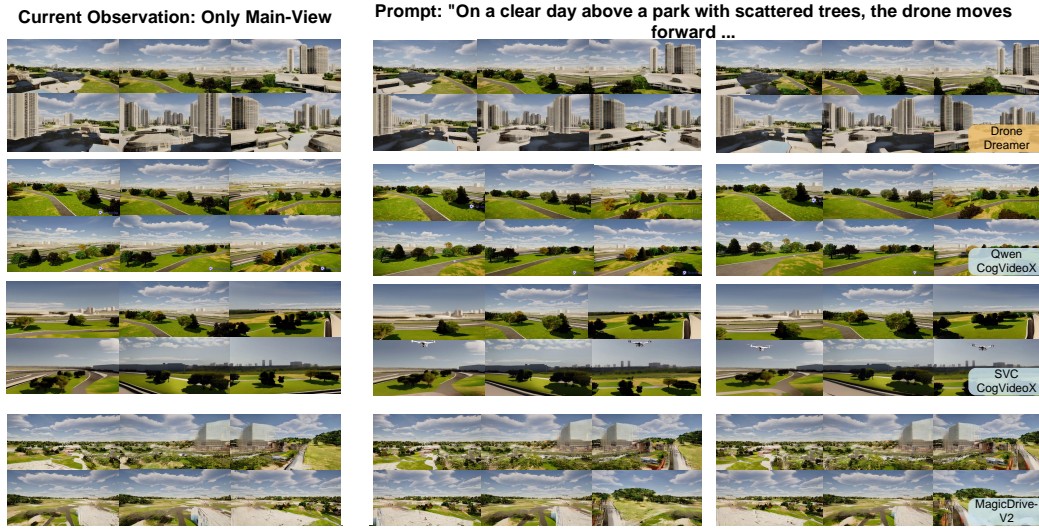

Figure 5: Qualitative comparison between DroneDreamer and baseline methods on multi-view video generation.

### 4.2.2 QUANTITATIVE RESULTS

Table 1 presents the experimental results of our method compared to the baseline models. We consider two groups of comparisons, based on which we draw the following conclusions.

**Our proposed DroneDreamer achieves the best overall performance.** Compared to the state-of-the-art methods, we have achieved the best balanced performance in terms of image generation quality and video consistency. Compared to the end-to-end model, we improved by 57.6% in the FID metric and 69.1% in the FVD metric. In terms of scene control ability, we achieved the best results in the mIoU metric, with a 58.6% improvement compared to the existing best method. Compared to the composite model, we achieved comparable results in the FID metric, particularly in real-world environments, where the gap is minimal. However, in terms of FVD, we achieved an improvement 63.1%. Notably, in complex flight environments, DroneDreamer shows a significant improvement compared to the best model.

**Baseline models exhibit poor viewpoint consistency and weak motion coherence.** Existing state-of-the-art end-to-end models and composite models built with image-to-video generation perform poorly in terms of viewpoint consistency, as indicated by high FVD scores and low mIoU scores. The high image generation quality in these models is reflected by low FID scores but high FVD scores, indicating limited variation between video frames and a lack of overall motion and spatial connection between multi-view videos. The original baseline model exhibits increasingly pronounced multi-view inconsistency as flight duration increases, as shown in Figure 5. This highlights that current image-to-video models prioritize image quality but still have room for improvement in real-world environmental perception, and are far from achieving general world model capabilities.

### 4.3 PROGRESSIVE TRAINING RESULTS

This section provides an analysis of the progressive training results, highlighting the performance improvements at each stage and validating the effectiveness of the proposed training strategy. The detailed results can be found in Table 2. At each stage of training, we observe consistent performance improvements. In the final stage, while there is a slight decline in the visual quality of simulated images, substantial gains are achieved in video generation quality and the realism of style-transferred images.

| Stage | Hybrid Styles | | Realistic Styles | | Simulated Styles | |
|---|---|---|---|---|---|---|
| | FID↓ | FVD↓ | FID↓ | FVD↓ | FID↓ | FVD↓ |
| Magicdrive-V2 | 173.02 | 1500.62 | 230.06 | 1619.90 | 172.51 | 1524.40 |
| Stage1 | 106.03 | 658.70 | 145.01 | 707.11 | 121.11 | 697.56 |
| Stage2 | 94.85 | 752.49 | 104.99 | 754.23 | **121.08** | 699.76 |
| **Stage3(ours)** | **73.31** | **463.94** | **83.62** | **516.21** | 138.53 | **493.84** |

Table 2: Quantitative progressive training results showing the performance gains across different training stages.

## 5 RELATED WORK

### 5.1 WORLD MODEL

World models represent the vision for general artificial intelligence, simulating real environments, supporting agent-environment interactions, and providing advanced reasoning and predictions for decision-making (LeCun, 2022; Ha & Schmidhuber, 2018; NVIDIA, 2025b). Research on world models focuses on three areas: First, general world models aim to create scalable representations using large datasets for understanding and predicting complex environments (Assran et al., 2025; Gu et al., 2024; Bruce et al., 2024). Second, edge-based models leverage world models for predictive perception, training agents in interactive environments (Wu et al., 2024; Zhou et al., 2024a) or supporting decision-making through predictions (Hu et al., 2025; Assran et al., 2025). Third, autonomous driving models simulate traffic scenarios and provide predictions for safer driving (Ruiyuan Gao, 2025; Wu et al., 2025; NVIDIA, 2025a). However, low-altitude environments, which offer more freedom and complex spatial distributions, have yet to be explored in depth (Gao et al., 2024a), despite their potential for evolving into generalized world models.

### 5.2 SIM-TO-REAL DOMAIN ADAPTATION

Sim-to-Real domain adaptation reduces the distribution gap between simulated and real-world data, crucial for tasks with data scarcity, high annotation costs, and diverse real-world scenarios. Classical methods rely on large-scale datasets from both domains, while recent work focuses on few-shot or zero-shot adaptation. Existing techniques are applied mainly in two areas: artistic style transfer, where the source image's style is adapted to match a reference image (Wang et al., 2025; Xu et al., 2024), and semantic segmentation, where models trained on synthetic data are applied to real-world data (Chigot et al., 2025; Hoyer et al., 2022). However, these methods are mainly used for data processing and have not been integrated with video generation. Combining them with diffusion models can reduce parameter counts and unlock their full potential.

## 6 CONCLUSION

This paper introduces the first low-altitude multi-view world model, capable of predicting future multi-view low-altitude flight videos based on current observations. This paper introduces a multi-view low-altitude video dataset containing 365 scenes, which facilitates the training and evaluation of world models. We propose two simple yet effective improvements and design a three-stage training plan. Experimental results demonstrate that our proposed DroneDreamer achieves superior performance in terms of controllability prediction and multi-view consistency. For future work, we plan to integrate the model with autonomous driving models and street scene generation models, forming a multi-scale world model.

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

# A APPENDIX

## A.1 LLM USAGE

In this work, we employ large language models (LLMs) as writing assistants to facilitate English composition and to improve the linguistic quality of the manuscript.

## A.2 LOW-ALTITUDE UAV FLIGHT VIDEO DATASET

In this section, we present the detailed construction pipeline of our dataset and provide information about its composition. We constructed a multi-view UAV flight dataset containing 365 urban scenes, along with corresponding trajectory, text, and semantic segmentation annotations, with representative cases shown in the appendix A.8.

**Dataset Construction Pipeline** Our multi-view UAV flight data is sourced from a simulated urban environment created in our lab, covering various flight trajectories, altitudes, and maneuvers. We employed large multimodal models to annotate the flight environment and motion poses of the UAV. To enhance data style, we used the EoMT model (Kerssies et al., 2025) for semantic segmentation and applied the CACTIF model (Chigot et al., 2025) to transfer the style from real-world UAV flight data to some simulated data, ensuring realistic style adaptation, as shown in Figure A.7. Finally, we performed over 100 hours of manual refinement, filtering out scenes not included in the Cityscapes (Cordts et al., 2016) catalog, such as rivers, mountains, and other irrelevant environments.

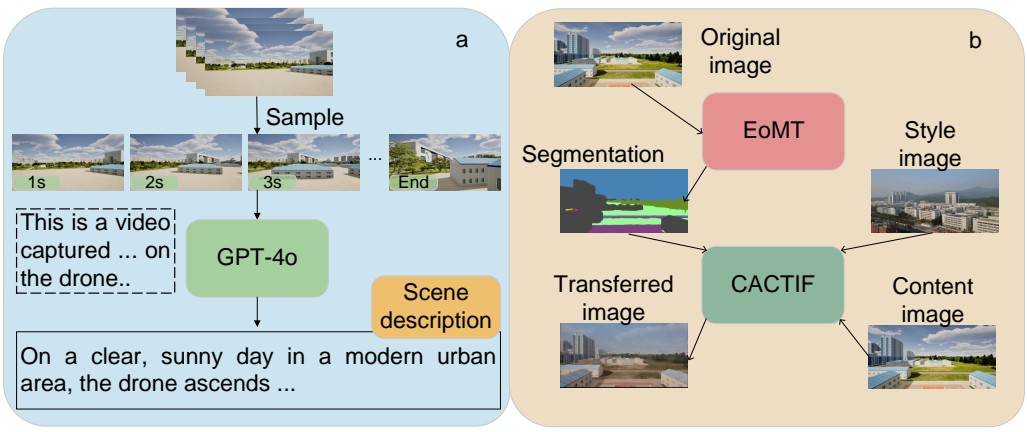

Figure A.6: a. Pipeline for Dataset Scene Annotation. b. Pipeline for Dataset Image Style Transfer

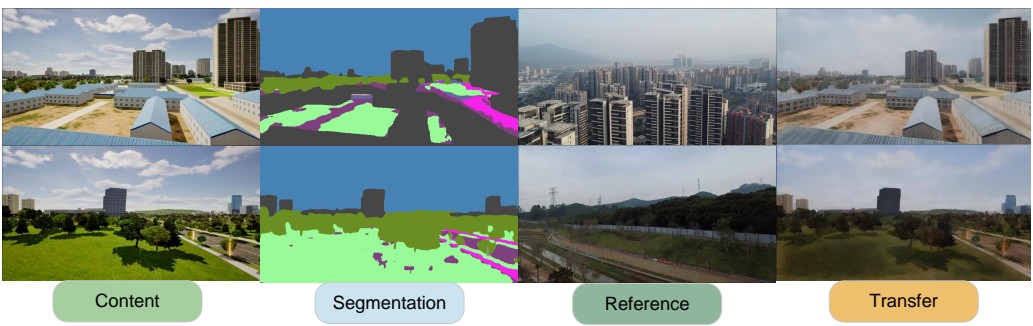

Figure A.7: Examples of Image Transformation.

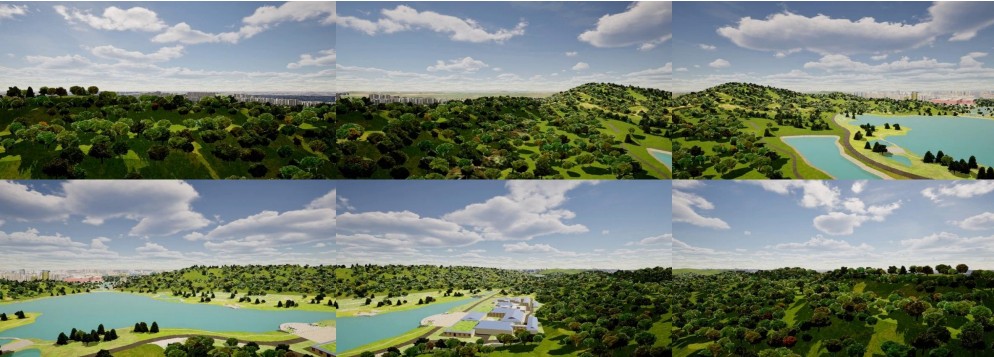

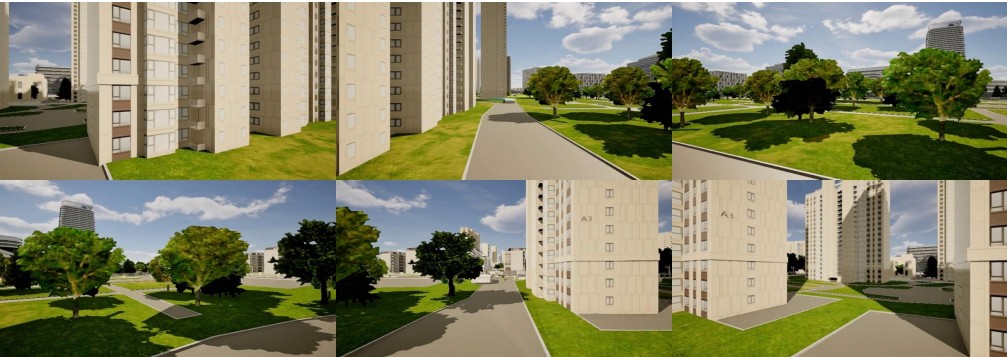

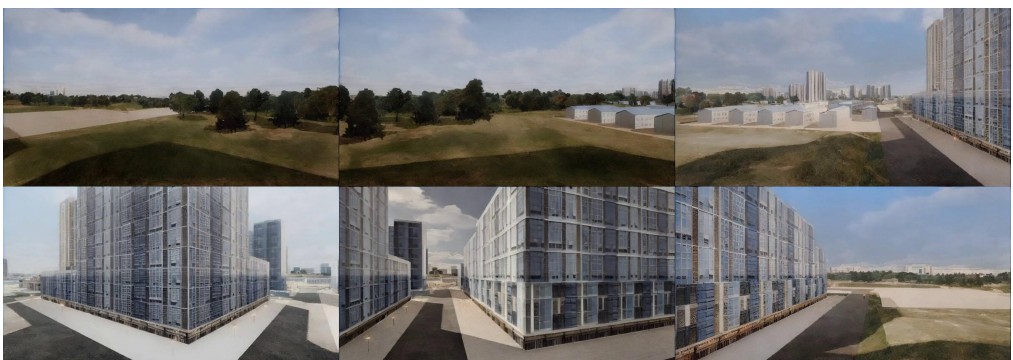

Figure A.8: From top to bottom, we show suburban scenes, urban scenes, and scenes after realistic style transfer.

**Dataset Statistics** Our low-altitude UAV flight dataset contains a total of 365 scenes, of which 310 are used for training, including 303 simulated images and 7 realistic-style images. The remaining 55 scenes are reserved for testing, consisting of 54 simulated images and 1 realistic-style image. The average scene length is 207 frames, with a median of 180 frames. Additional image examples are presented below, as shown in Figure A.8.

## A.3 ADDITIONAL EXPERIMENTAL DETAILS

### A.3.1 MODEL CONFIGURATION

We use the 3D VAE framework from CogVideoX and the trained diffusion model to generate a 17-frame street scene video at a resolution of 424×800 under the guidance of the first-frame main view.

We used 53 17-frame clips as the validation set, with 34 clips from the simulated dataset and 19 clips from the style-transformed dataset. In the composite model validation, we first use the first

model, referencing the first-frame main view, to generate the first-frame multi-view. Then, we use the second model to generate the video for multiple viewpoints.

### A.3.2 SUPPLEMENTARY RESULTS

**Comparison of Parameters and Inference Time** We compare our model with the combined model in terms of the number of parameters and inference time. For the inference of a six-view 49-frame video using 30 denoising steps, the parameter counts and inference times are reported in Table A.3. Compared with baseline models, our model achieves low-altitude multi-view flight video inference with reduced parameter count and lower inference time.

| Model | Parameters | Inference Time |
|---|---|---|
| Qwen+CogVideoX-i2v-5B | LLM+5B | (3+2.5)min |
| SVC+CogVideoX-i2v-5B | 1.3B+5B | (2+2.5)min |
| **DroneDreamer** | **1.9B** | **2.0min** |

Table A.3: Comparison of Model Parameters and Inference Time.

