# OpenReview forum: "DroneDreamer: Multi-View Low-Altitude World Model with Adaptive Control"
_ICLR.cc/2026/Conference — Submitted to ICLR 2026_

### Official Review · Reviewer_KEFu · 2025-10-26

**Soundness:** 2
**Presentation:** 3
**Contribution:** 2
**Rating:** 4
**Confidence:** 2

**Summary:**

This propose DroneDreamer, a Low-Altitude World Models(LAWM) that incorporates an adaptive viewpoint control mechanism and an image style domain adaptation technique, enabling multi-view conditional generation with limited conditions. Experiments show the proposed method is better than the baseline models.

**Strengths:**

1. This paper proposes the first (according to the paper claimed) low-altitude multi-view world model, to predict trajectory-consistent multi-view flight videos based on the current front-view observation.
2. Experiments show the proposed method is better than the baseline models.

**Weaknesses:**

1. This paper might deal with a new research direction, but all the used technical components seem just coming from existing works and thus with limited technical contribution.
2. Although experiments show the proposed method is better than the baseline models, but the baseline methods seems designed by this paper and too simple to be compared with. Furthermore, according to the figures (qualitative resutls) of this paper, it is also difficult to understand how the proposed method is better than other methods.
3. This paper claims it applies style adaption to reduce the domain gap between simulation data and real-world environments. But, according to the experiments, I can't find any real-world results. The results look still simulation style.

**Questions:**

1. It would be better to figure out what's the new design by this paper.
2. Current results are difficult to understand how good the proposed method is. Maybe, a demo video to show the comparison result would be better.

---

### Official Review · Reviewer_kySR · 2025-10-29

**Soundness:** 2
**Presentation:** 1
**Contribution:** 2
**Rating:** 2
**Confidence:** 5

**Summary:**

This paper presents DroneDreamer, the first world model developed for various downstream tasks based on observations captured from multi-view, low-altitude flight scenarios. The proposed model adopts a Diffusion Transformer (DiT) architecture integrated with a 3D Variational Autoencoder (3D VAE), and includes an adaptive control mechanism capable of leveraging multiple control inputs such as text, viewpoint, and trajectory. To further enhance model controllability and overall performance, the authors construct a new multi-UAV dataset comprising 365 scenes and employ a three-stage training pipeline. Experimental results show that DroneDreamer achieves comparable visual quality and consistency to existing baselines in the novel-view video generation task, outperforming end-to-end models and performing on par with composite models.

**Strengths:**

- Developing a new world model tailored for low-altitude, aerial-view scenarios is a valuable direction. Such efforts are especially welcomed if the model’s applicability is convincingly demonstrated across multiple downstream tasks.

- Regardless of the degree of technical novelty, the introduction of a guide-control module appears to be an appropriate and practical approach to manage diverse control parameters originating from various guidance types, including text, viewpoint, and trajectory.

- The creation of a new dataset is also an appreciated contribution. If the paper provides well-designed experiments validating its unique properties and usefulness beyond data collection, this addition could be strongly recognized by the community.

- The adoption of a specialized multi-stage training strategy for the proposed world model is an interesting and meaningful design choice, contributing to improved controllability and overall model performance.

**Weaknesses:**

**Insufficient Empirical Support for Claimed Contributions**

The main concern is that the proposed contributions are not well supported in the manuscript.

- To verify the impact of developing a world model, it is crucial to demonstrate its applicability across multiple downstream tasks. However, the paper only evaluates the approach on a video generation task, which limits the generality of its contribution.

- Although a new dataset is introduced, no representative images or examples illustrating its distinct properties or advantages are presented in the paper.

- If the paper aims to validate the three-stage training strategy, it should include a comparison with a single end-to-end training strategy in Table 2. Currently, it only compares individual stages and baseline methods.

**Lack of Clarity and Inconsistent Presentation**

The presentation of technical details is often unclear, making it difficult to follow the model design and methodology.

- It is not explicitly described how each type of control parameter (text, viewpoint, trajectory, etc.) is processed within the control blocks.

- Figure 3 is difficult to interpret. The origin and destination of the arrows between control blocks and base blocks are not specified, and the meaning of “seq comb. emb” is undefined.

- Section 3.3 is hard to follow. The process of reference image injection and the purpose of the masking mechanism are not clearly explained. Is the mask only indicating the injection region, or does it serve another role?

- The meaning of “35” in Equation (1) is not defined.

- Some notations are missing definitions, such as “ov” and “mv” in Equations (2) and (3).

- Section 3.5 is also unclear, particularly regarding the second stage of the training pipeline — the paper does not specify where the new fully connected layer is inserted.

- In Figure 5, the qualitative results are difficult to interpret. How can frame consistency be visually assessed? What are the differences between results with and without prompts, and between the two prompt-based columns?

- In Table 1, the meanings of “hybrid,” “realistic,” and “simulated” styles are not explained, making comparison difficult.

**Limited Experimental Superiority**

The experimental results do not convincingly demonstrate the advantages of the proposed method.

- In Table 1, DroneDreamer’s performance is not superior to competing methods and is worse than some approaches in terms of FID.

- Additionally, the qualitative results in Figure 5 do not clearly show any distinct advantage or improvement over the baseline methods.

**Questions:**

Please address my concerns listed in the weakness section.

**Details Of Ethics Concerns:**

I don't have any ethical concern.

---

### Official Review · Reviewer_i2ci · 2025-10-30

**Soundness:** 2
**Presentation:** 1
**Contribution:** 1
**Rating:** 0
**Confidence:** 5

**Summary:**

This paper introduces DroneDreamer, a novel low-altitude world model (LAWM) designed to generate multi-view drone flight videos conditioned on limited input observations. The authors identify key challenges in applying world models to low-altitude UAV scenarios, including data scarcity, difficulty in obtaining control conditions, and domain gaps between simulated and real-world data. To address these, they propose an adaptive masking mechanism for conditional generation, a style adaptation module for sim-to-real transfer, and a three-stage progressive training strategy. The model is evaluated on a custom-built multi-view drone dataset and demonstrates significant improvements in multi-view consistency and controllability over strong baselines.

**Strengths:**

1. The authors present a holistic solution, including dataset construction, adaptive masking for control, style adaptation via cross-attention and AdaIN, and a progressive training strategy.
2. The model is rigorously evaluated against both end-to-end and composite baselines using multiple metrics (FID, mIoU, FVD) across different styles.

**Weaknesses:**

1. The core technical building blocks are all established methods: Diffusion models, cross-attention for conditioning, and AdaIN for style transfer. The adaptive masking mechanism, while practical, is also a straightforward extension of existing inpainting or conditioning techniques.
2. The dataset is predominantly synthetic, with only a small fraction of real-world style-transformed images. While style adaptation is proposed, the evaluation does not sufficiently demonstrate generalization to real-world flight data, which is critical for low-altitude UAV applications.
3. The baselines (MagicDrive-V2, Qwen+CogVideoX, SVC+CogVideoX) are not all originally designed for low-altitude settings. A comparison with other recent world models (e.g., Genie, V-JEPA) or UAV-specific models would strengthen the claims of superiority.
4. The paper focuses on generative metrics but does not evaluate the model’s utility in downstream tasks such as navigation, planning, or prediction, which are central to the concept of a "world model."
5. The paper does not quantitatively disentangle the individual contribution of each proposed component (e.g., the adaptive masking mechanism, the style adaptation module, the main-view cross-attention). It remains unclear how much each element contributes to the final performance gains, making it difficult to assess their necessity and relative importance.

**Questions:**

1. The core of DroneDreamer integrates several existing techniques (adaptive masking, cross-attention, AdaIN). Could you provide ablation studies to quantitatively dissect the individual contribution of each component to the final performance?
2.The paper positions DroneDreamer as a "world model." However, the evaluation is primarily based on generative metrics (FID, FVD). How do you envision or demonstrate its utility for fundamental world model tasks like planning, prediction, or serving as an internal simulator for a drone agent? Have you tested its performance in such downstream tasks?
3. The model is trained predominantly on simulated data with a very small set of style-adapted real images. Can you provide evidence of its performance on completely unseen, real-world drone footage that was not used in the style transfer process? Does the model maintain multi-view consistency and controllability in such a challenging, out-of-distribution setting?
4. Could you elaborate on the details of the three-stage training strategy? Specifically, how did you prevent catastrophic forgetting of the base model's capabilities when freezing parameters in stage 3?

---

### Official Review · Reviewer_qKBc · 2025-11-03

**Soundness:** 2
**Presentation:** 2
**Contribution:** 2
**Rating:** 4
**Confidence:** 4

**Summary:**

The paper introduces low-altitude world models emulating flighing conditions for UAVs operating near ground level that must reason over six synced viewpoints. The authors argue that existing multi‑view world models from autonomous driving/robotics are not directly applicable due to scarce aerial data, missing/expensive control signals (such as BEV maps, 3D boxes), and a sim‑to‑real gap. Therefore, they introduce DroneDreamer an architecture that extends a 3D‑VAE + DiT backbone (from MagicDrive‑V2) with adaptive masking at the input level and style adaptation via main-view cross-attention alyer and semantics-conditioned AdaIN to narrow the sim-to-real distribution gap. Also a three-stage training curriculum is used. S1 involves using simulated multi‑view flights without control. S2 adds main-view reference control, while S3 freeze base weights and finetune style layers on a small set of style‑transferred samples with semantic labels.

This paper also introduces a new six‑view low‑altitude UAV dataset with 365 scenes, collected in a simulated environment and augmented (using LMMs) via semantic segmentation + image style transfer to emulate real imagery.

Experiments are conducted on "hybrid / realistic / simulated” style splits. The method improves FVD (video consistency/motion) and mIoU (controllability proxy) compared to an end‑to‑end baseline (MagicDrive‑V2) and two “composite” pipelines (SVC+CogVideoX and Qwen‑Image‑Edit+CogVideoX‑I2V). FID (which measures image realism) is roughly comparable to composites on realistic data but still worse on simulated/hybrid.

**Strengths:**

* The input adaptive masking for reference-frame control is a simple trick that can stabilize sampling even when structured controals are unavailable, which is realistic for UAVs.
* Semantics-conditions AdaIN and main-view cross-attenion inside the model is a useful engineering choice for style adaptation without huge overheads.
* The three‑stage scheme helps the model learn generation first, then control, then style transfer without catastrophic degradation.

**Weaknesses:**

* It is not clear how many reference frames are used during training/inference (only the first main view? any subset?), or even how the mask is scheduled temporally across views.
* Another ambiguous part is the way in which control signal (camera intrinsics/extrinsics) is injected into attention.
* The authors sometimes interpret FID differences as "comparable" despite large gaps (see Table 1 - the bolded values are all over the place). Qwen+CogVideoX-i2v-5B has better FID numbers than DroneDreamer even though the bolded values are for the proposed method.
* (Minor) Page 4 has many typographical errors when rendering mathematical formulas (they get glued to the text).
* There is no ablation quantifying the independent contribution for masking, main‑view cross‑attention or class‑wise AdaIN.
* In Table 2, Stage‑2 FVD on hybrid/realistic worsens vs Stage‑1 before improving in Stage‑3. The paper does not analyze why (overfitting to main‑view control? training instability after adding new FC layer?). Clarification is needed.
* The real-world generalization is largely untested due to only 7 style‑transferred “realistic” scenes in training and 1 in test (from A.2)
* Not sure what altitudes does the dataset emulate. The camera layout is fixed (six evenly spaced views) and it is unclear if the method handles varying camera rigs or changing altitudes outside the training distribution.
* Composite pipelines (SVC/Qwen + CogVideoX‑I2V) are strong on FID but weak on multi‑view coherence. However, the paper could strengthen fairness by tuning composites for multi‑view.
* Inference time (“2.0 min for six‑view 49‑frame, 30 DDIM steps”) is reported without hardware details making comparisons hard to interpret (Table A.3).

**Questions:**

1) How many reference frames are injected during training/inference and how is the mask aligned temporally across views?
2) Why does FVD degrade from Stage‑1 → Stage‑2 (Table 2)?
3) Do results hold on real, handheld UAV videos not used for style transfer (no simulator content)? Even a small “in‑the‑wild” test would materially strengthen claims.
4) What happens if composite baselines are given explicit cross‑view camera poses or a view‑consistency loss? This could test whether DroneDreamer’s gains are architectural or mainly due to conditioning availability.
5) What are the main causes of failure? Please describe the current method limitations.
6) Any ablations?

---

### Meta-Review · Area_Chair_Ma4d · 2025-12-26

**Summary:**

This paper presents a world model approach for drone. Yet, as all reviewer unanimously agree, the paper has not been ready for publish. The authors did not respond to the reviewers' comment as well. My recommendation is rejection.

**Reviewer Concerns:**

The major concern is the novelty regarding the method. It seems to be a reorganization of many existing approach to a new field.

**Reviewer Scores:**

The paper receives an initial score of 4 0 2 4. No reviewer changed their decision as there is no response from the authors.

---

### Decision · Program_Chairs · 2026-01-26

Reject